# Hexagonal Closed-Packed Precipitation Enhancement in a NbTiHfZr Refractory High-Entropy Alloy

**Yueli Ma [1], Shiwei Wu [1], Yuefei Jia [1]**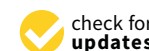**, Pengfei Hu [1], Yeqiang Bu [2], Xiangru Chen [3],\*, Gang Wang [1,3],\*, Jiabin Liu [2], Hongtao Wang [2] and Qijie Zhai [3]**

[1] Laboratory for Microstructures, Institute of Materials, Shanghai University, Shanghai 200444, China; mayl0212@163.com (Y.M.); wushiwei@shu.edu.cn (S.W.); JiaYF710@163.com (Y.J.); hpf-hqx@shu.edu.cn (P.H.)
[2] School of Materials Science and Engineer, Zhejiang University, Hangzhou 310027, China; yeqiangbu@zju.edu.cn (Y.B.); liujiabin@zju.edu.cn (J.L.); htw@zju.edu.cn (H.W.)
[3] Center for Advanced Solidification Technology (CAST), School of Materials Science and Engineering, Shanghai University, Shanghai 200444, China; qjzhai@shu.edu.cn
\* Correspondence: cxr16@shu.edu.cn (X.C.); g.wang@shu.edu.cn (G.W.); Tel.: +86-21-6613-5028 (X.C.)

**Abstract:** A NbTiHfZr high-entropy alloy (HEA) with a main phase of body-centered cubic structure is fabricated. Some hexagonal closed-packed (hcp) precipitates are observed in this alloy. A thermal-mechanical process, i.e., cold-rolling followed by annealing, can manipulate the volume fraction of the hcp nano-precipitates that can enhance strength and ductility. The enhancement is tailorable as a function of the volume fraction of the hcp nano-precipitate. The results indicate that the strength-ductility property can be manipulated via adjusting post-deformation heat-treatment methods, which provide a new strategy by utilizing metastability at high-temperature to design high strength refractory HEAs (RHEAs) without lost in ductility.

**Keywords:** high-entropy alloy; cold-rolling; annealing; ductility; nano-precipitation particle

## 1. Introduction

Refractory alloys, a group of materials with the melting temperatures ($T_m$)>2123 K, typically exhibit an excellent high-temperature strength, a good creep resistance and an excellent superconductivity [1–12]. Recently, high-entropy alloys (HEAs) based on the refractory elements have been developed, which are potential alloys to be used at high temperatures in future [1–7]. For instance, some new refractory HEAs (RHEAs) with a body-centered cubic (bcc) structure, such as $Ti_{20}Zr_{20}Hf_{20}Nb_{20}V_{20}$ (at.%) and $Ti_{20}Zr_{20}Hf_{20}Nb_{20}Cr_{20}$ HEAs were successfully developed [13]. A $Ti_{20}Zr_{20}Hf_{20}Nb_{20}Ta_{20}$ HEA [4] shows superior mechanical properties in compression at high temperature, as compared to conventional high-temperature alloys. Nevertheless, the bcc HEAs developed up to date suffer from poor ductility at room temperature, which makes them difficult to be processed, and thus limits their applications [2,14–17]. For example, Wu et al. [16] demonstrated firstly room temperature tensile plasticity in a $Nb_{25}Ti_{25}Hf_{25}Zr_{25}$ RHEA that exhibited the yield strength of approximately 879 MPa accompanied by a room temperature plasticity of 15%. Dirras et al. [17] reported the tensile yield strength of the as-cast TiZrHfTaNb HEA is 800–840 MPa, but the fracture strain is only ~4%. Accordingly, intensive efforts have been devoted to improve the ductility of bcc HEAs at room temperature [18–22]. Recently, Huang et al. enhanced the comprehensive properties of brittle TaHfZrTi RHEAs through generating a dual-phase structure [22]. Sheikh et al. proposed an alloy design strategy to intrinsically make HEAs ductile based on the electron theory and more specifically to decrease the number of valence electrons through alloying [19]. However, how to

make these RHEAs with good ductility up to bulk engineering materials through simple processes for industrial applications, remains a mystery.

In the present study, through cold-rolling followed by annealing, we introduce many hexagonal close-packed (hcp) nano-particles precipitation to improve the ductility of a bcc RHEA. A refractory $Nb_{25}Ti_{25}Hf_{25}Zr_{25}$ HEA with a combination of high tensile strength and plasticity is presented. The $Nb_{25}Ti_{25}Hf_{25}Zr_{25}$ RHEA has a bcc structure with the lattice parameter a = 345 pm and contains a small amount of hcp phase. The structural stability, deformation and fracture behavior of this RHEA are studied. The strengthening mechanism for the excellent tensile plasticity of this RHEA is discussed.

## 2. Materials and Methods

The master alloy of the refractory $Nb_{25}Ti_{25}Hf_{25}Zr_{25}$ HEA was prepared by arc-melting a mixture of pure metals (purity $\geq$ 99.95%), and was remelted at least five times to promote the chemical homogeneity in a Ti-gettered high-purity argon atmosphere. The ingots were 11.5 mm in thickness and 28.4 mm in diameter. After that, the melted ingots were poured into a metallic mould to form a sheet with a thickness of 9 mm, a length of 100 mm and a width of 40 mm. The sheet was step-by-step cold-rolled along the length direction at room temperature by using a laboratory scale rolling equipment having rollers with a diameter of 16 mm, and a length of 260 mm. The surfaces of the sample and rollers were lubricated using oil in each step, and the reduction in thickness by each step was ~0.2 mm. The thickness of the sheet was reduced to 2 mm that is approximately 77.8% reduction. After rolling, the surface of the final rolled sheet was smooth and with no obvious cracks (see Supplemental materials Figure S1). After cold-rolling, the RHEA was annealed for 1 h at 1073 K and 1273 K in vacuum followed by furnace cooling to room temperature. Tension tests with a strain rate of $1 \times 10^{-3} \cdot s^{-1}$ were performed with an MTS 653 machine at room temperature. The test samples had a dog-bone shape with a gauge geometry of 18 mm × 2 mm × 4 mm. The microstructure and fracture morphologies of the samples were investigated with a scanning electron microscope (SEM) (SU-1510, Hitachi Company, Tokyo, Japan). After mechanically grinding, the SEM samples were electrochemically polished in a mixture solution at the temperature of −30 °C. The mixture solution was composed of perchloric acid (in volume fraction of 6%), n-butyl alcohol (in volume fraction of 34%) and methyl alcohol (in volume fraction of 60%). Phase analysis was performed in a Rigaku D\max-2550 X-ray diffractometer (XRD, Rigaku Company, Tokyo, Japan) with a Cu-Kα radiation. The XRD samples were mechanically polished on the diamond paper with the particle size of 5 μm. Electron back scattering-diffraction (EBSD) tests were performed in a CamScan Apollo 300 scanning electron microscopy (SEM, CamScan Company, Waterbeach, UK) equipped with a HKL Technology EBSD system. After fracture, the area close to the fracture surface were cut for transmission electron microscope (TEM) observation, which was ground to 30 μm in thickness, and then twin-jet polished in a 5% (vol%) perchloric acid alcohol solution at a temperature of 243 K to generate an electron transparent region. The TEM observation was carried out in a JEM-2010F microscope operated at 200 kV (TEM, JEOL Company, Tokyo, Japan).

## 3. Results

### 3.1. Structural Evolution

Figure 1a shows the XRD patterns of the as-cast RHEA, and the cold-rolled RHEAs followed by annealing at 1073 K and 1273 K. Peaks of a bcc phase with a lattice parameter a = 3.45 Å are observed, which does not show any phase separation but a bcc single-phase microstructure (Figure 1a). The enlarged peaks in the 2θ ranged from 25° to 40° (Figure 1b) show that the two small peaks are located at 33.44° and 38.83°, respectively, in the 1073 K-annealed RHEA, which point to a hcp phase. Further enlarging the peaks in the 2θ range from 50° to 60° (Figure 1c) also show one small diffraction peak at 56.05° pointing to the hcp phase. Around the peak located at 90.57°, a small diffraction peak appears in the 1273 K-annealed RHEA (Figure 1d), which also indicates the hcp phase. The weak

diffraction peaks indicate that the volume fraction of the hcp phase is very small. For the as-cast RHEA, there is no any diffraction peaks indicating the hcp phase.

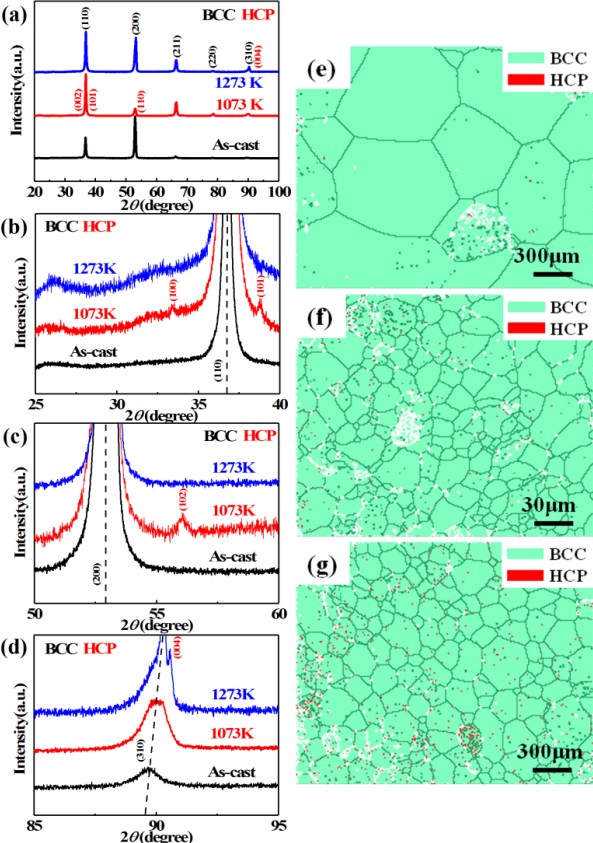

**Figure 1.** Phase structure of the $Nb_{25}Ti_{25}Hf_{25}Zr_{25}$ refractory high entropy alloy (RHEA). (**a**) XRD patterns of the RHEAs in as-cast, cold rolled-and-annealed at 1073 K and 1273 K states; (**b–d**) Enlarged XRD patterns of the RHEAs at different states; (**e**) Electron back scattering-diffraction (EBSD)image of the as-cast RHEA; (**f**) EBSD image of cold-rolled RHEA followed annealing at 1073 K; (**g**) EBSD image of cold-rolled RHEA followed annealing at 1273 K. The white area is the one without Kikuchi patterns during EBSD scanning which cannot be determined in EBSD-data analyzing software. The occurrence of such area is due to residual stress.

From the EBSD images (Figure 1e–g), the as-cast RHEA mainly consisted of equiaxed grains (Figure 1e). The grain size was measured to be ~350 ± 50 μm according to the EBSD images through a line-intersection method. After annealing at 1073 K, the cold-rolled structure was completely recrystallized and the average grain size was 10 ± 5 μm (Figure 1f). Annealing at 1273 K makes the grain grow up to 140 ± 10 μm. From Figure 1f,g, it can be seen that the grain size is not uniform whether the cold-rolled RHEA is annealed at 1073 or 1273 K. The microstructure of the RHEA is changed by annealing, and the grain size is refined.

To further confirm the phase structure from the XRD patterns, the EBSD images are shown in Figure 1. For the as-cast RHEA, although a few red spots pointing to the hcp phase are shown in Figure 1e, the volume fraction of the red spots is too small. Thus, considering the XRD pattern, the hcp phase does not precipitate in the as-cast RHEA. After cold-rolling accompanied with annealing at 1073 K, besides the reduction in the grain size, the amount of the red spots, i.e., the hcp phase increases significantly (Figure 1f), which never appears in the relevant RHEAs [16,23]. Annealing at 1273 K, more hcp phases precipitate. Although the quantitative estimation of the volume fraction of the hcp phase is not possible based on the EBSD images, it can be deduced that the amount of the hcp phase

increases after cold-rolling and annealing (see Supplemental materials Figure S2). The cell parameters of the hcp phase measured from the XRD patterns are listed in Table 1.

**Table 1.** Hcp precipitates and tensile properties of the 1073 and 1273 K-annealed RHEAs.

| Condition | hcp Precipitates | | | | | | | | Tensile Properties | | |
|---|---|---|---|---|---|---|---|---|---|---|---|
| | Composition (at.%) | | | | Lattice Constant (Å) | | | | $\sigma_{0.2}$ (MPa) | $\sigma_{UTS}$ (MPa) | $\varepsilon_u$ (%) |
| | Hf | Nb | Ti | Zr | $a_{HRTEM}$ | $c_{HRTEM}$ | $a_{XRD}$ | $c_{XRD}$ | | | |
| As-cast | – | – | – | – | – | – | – | – | 636 ± 13 | 652 ± 13 | 7.5 ± 0.5 |
| 1073 K | 32.9 ± 2.9 | 22.1 ± 1.9 | 16.2 ± 2.2 | 28.9 ± 1.1 | 2.73 ± 0.03 | 5.27 ± 0.03 | 3.16 ± 0.11 | 4.26 ± 0.32 | 778 ± 3 | 934 ± 27 | 21.4 ± 2.7 |
| 1273 K | 33.2 ± 2.1 | 21.9 ± 1.4 | 16.3 ± 1.5 | 29.0 ± 0.8 | 2.75 ± 0.04 | 5.24 ± 0.02 | 3.05 ± 0.14 | 4.34 ± 0.00 | 769 ± 71 | 941 ± 117 | 27.4 ± 1.8 |

The subscripts 'HRTEM' and 'XRD' represent the lattice constants which were calculated via using high-resolution transmission electron microscopy patterns (HRTEM) and XRD peaks, respectively.

From the EBSD results, we can see that the amount of the hcp phase after annealing is obviously increased. In addition, a large fraction of low-angle grain boundaries (2–15°) is observed according to the EBSD. Figure 2 presents a high-angle annular dark field scanning transmission electron microscopy (HAADF-STEM) image and energy-dispersive spectroscopy (EDS) maps of the RHEA after cold rolling and annealing treatments, which demonstrates that distinct segregations of Hf and Zr exist in the RHEA. The atomic percentages of the elements of Hf, Nb, Ti and Zr are 32.9 ± 2.9%, 22.1 ± 1.9%, 16.2 ± 2.2% and 28.9 ± 1.1%, respectively, obtained by point scanning (see Figure S3 in Supplemental materials), which further verifies that the hcp particles are rich in Hf and poor in Ti. In the present study, Nb is the element with the highest melting temperature (2750 K) followed by Hf (2506 K), Zr (2128 K) and Ti (1941 K). Hf is dominant within the second phase, implying that a thermodynamically driven phase separation occurs [24].

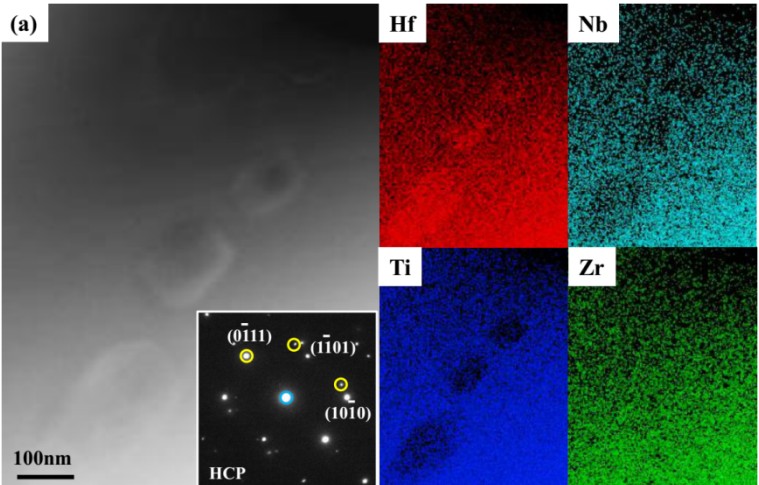

**Figure 2.** High-angle annular dark field scanning transmission electron microscopy (HAADF -STEM) image and energy-dispersive spectroscopy (EDS) maps of 1073 K-annealed RHEA. The illustration is a selective electron diffraction pattern of the hcp phase. The Hf-rich, (Ti,Nb)-depleted areas correspond to the hcp precipitates.

*3.2. Tensile Properties of the Cold-Rolled HEA*

Figure 3 shows the true stress–strain curves of the RHEAs. The corresponding mechanical parameters, i.e., yield strength, $\sigma_{0.2}$, ultimate tensile strength, $\sigma_{UTS}$, and fracture strain, $\varepsilon_f$, are summarized in Table 1. To exclude the occasional cases, the tension test for each sample was repeated for five times. From Table 1, it can be seen that for as-cast RHEA, the $\sigma_{0.2}$ and $\sigma_{UTS}$ values approach the maximum values of 636 ± 13 MPa and 652 ± 13 MPa, respectively. Annealing at 1273 K after rolling causes the $\varepsilon_f$ value to be increased to 27.4 ± 1.8%. It is obvious that cold rolling followed

by annealing at 1073 and 1273 K significantly improves the strength and ductility of the RHEA, as compared to the case of the as-cast RHEA.

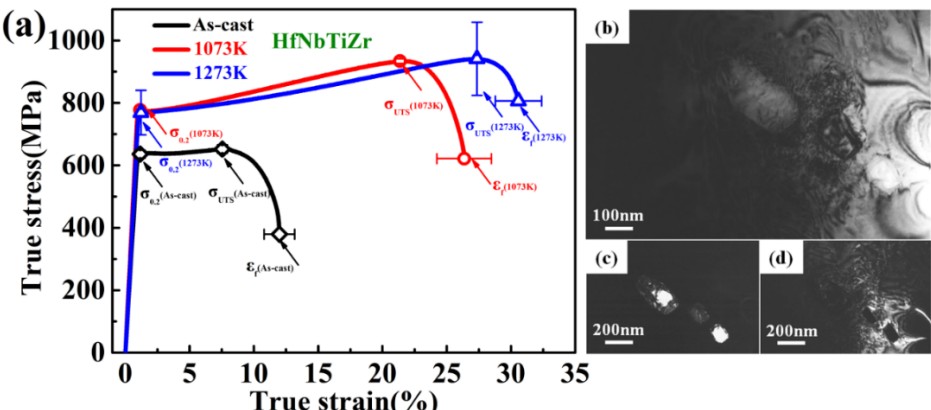

**Figure 3.** (**a**) Tensile true stress–strain curves of the as-cast, cold-rolled and annealed at 1073 and 1273 K RHEAs; (**b**–**d**) Bright-field (BF) and dark-field (DF) images of the hcp precipitates after tension.

Plastic deformation of the RHEA occurs at a modest strain-hardening rate. According to stress-strain curves, the strain-hardening rates ($d\sigma/d\varepsilon$) as functions of the true strain curves are plotted in Figure S4 in the Supplementary Materials. The strain-hardening rate curves of as-cast RHEA can be classified into three regions. The initial drop in the strain-hardening rate, which is observed in all the alloys, is in region I. It is known that continuously decreasing stain-hardening rate is correlated to a dislocation-slip-dominated deformation mechanism [25]. In region I, the strain-hardening rates of the as-cast RHEA monotonically decreasing with increasing strain from 19.3 to −2.1 GPa. After that, the strain-hardening rate of the as-cast RHEA gradually increases to a peak value of approximately 4.4 GPa at the strain of 5.9% in region II. The strain-hardening rate curve studied here is unique and is rarely reported in previous literatures [26–28]. Whereas, the change tendency of the strain-hardening rate curve is traditional in TWIP steels [29,30]. The strain-hardening rate progressively decreases with increasing strain for the as-cast and cold-rolled RHEAs, which is attributed to dislocation-slip-dominated plastic deformation mechanism [31].

The strain-hardening rate curves for the cold-rolled RHEAs annealed at 1073 K and 1273 K demonstrate five different deformation regions as depicted in Figure S4 in the Supplementary Materials. The strain-hardening rate show non-monotonous evolutions. For the RHEA annealed at 1073 K, a continuously decreasing strain-hardening rate is observed initially. Then, the strain-hardening rate increases gradually (region II) to the maximum of 9.7 GPa, and then decreases again to the value of 9 GPa at the strain of 11.9% (region III). Another increase and downward trends emerge finally in regions IV and V, respectively. For the RHEA annealed at 1273 K, the strain-hardening rates decrease lineally from 15.7 to 3.6 GPa in region I, and then increases to 7.4 GPa in region II. In region III, the strain-hardening rate slightly increases to 7.8 GPa. After that, with straining, the strain-hardening rate further increases to 8.7 GPa. Eventually, the strain-hardening rate decreases to 5.3 GPa at the rupture strain. It is noted that the predominant mechanism defines the shape of strain-hardening rate curve, rendering either a monotonous decreasing form or a non-monotonous shape with a plateau or hump [22]. The fractographies indicate that a ductile separation occurs on the fracture surface, as shown in Figure S5 in the Supplementary Materials.

### 3.3. Microstructure

The microstructure of the as-cast RHEA is investigated by TEM. The bright-field (BF) image in Figure 4a indicates the bcc phase in the as-cast RHEA. The corresponding selected area diffraction (SAED) pattern is shown in the inset of Figure 4a. The TEM results indicate that only the bcc single

phase can be observed, which is consistent with the results from the EBSD and XRD observations. The bcc phase contains no dislocation in the as-cast RHEA (Figure 4a).

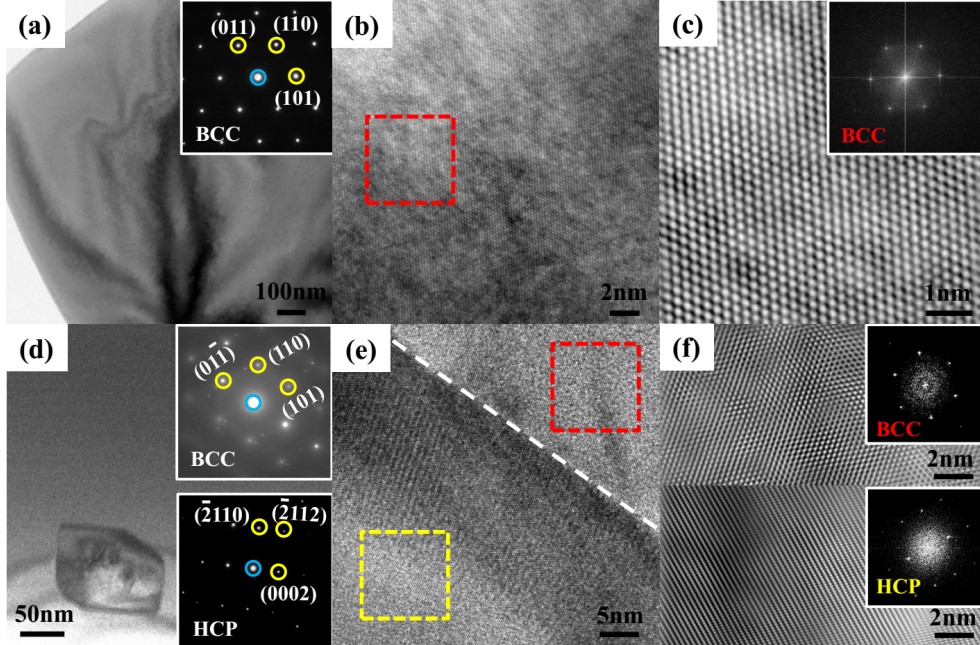

**Figure 4.** TEM results of as-cast and 1073 K-annealed RHEAs. (**a**) BF image of the as-cast RHEA without tension. The inset is SAED pattern; (**b**) HRTEM image of the as-cast RHEA; (**c**) Fast Fourier transformation (FFT) image of the area covered by red square in (**b**); (**d**) BF image of the hcp precipitate in the 1073 K-annealed RHEA without tension. The inset is the SAED patterns of the matrix and the hcp phase; (**e**) HRTEM image of the matrix and the hcp phase. Red and yellow squares represent the matrix and the hcp phase, respectively; (**f**) FFT images of the areas covered by squares in (**e**).

The microstructure of the cold-rolled RHEA annealed at 1073 K is shown in Figure 4d–f. Figure 4d shows the BF image, in which a dual-phase (bcc + hcp) structure appears. The corresponding SAED pattern shows that the matrix is the bcc phase and the second phase is the hcp phase. The average size of the hcp particles is measured by the line-intersection method from the TEM dark-field (DF) images. At least 10 images are measured to obtain the average value. The average sizes of the hcp phase particles in the 1073 K and 1273 K-annealed RHEAs are 153 ± 13 nm and 174 ± 4 nm, respectively (see Figure S6 in the Supplementary Materials). The number densities of 1073 K and 1273 K-annealed RHEAs are $8.3 \times 10^{11}/m^2$, and $9.3 \times 10^{11}/m^2$, respectively (see Figure S7). Compared with the 1073 K-annealed RHEA, both the number density and average precipitate size of the hcp phase increase when the annealing temperature is increased to 1273 K, indicating an enhancement of hcp precipitates occurs with increasing annealing temperature. After deformation, the hcp phase particles are surrounded by dense dislocation plug, the bright and dark-field images of hcp particles are shown in Figure S8b–d. It is obvious that the as-cast RHEA mainly contain the bcc phase. Annealing causes the hcp phase precipitating in the bcc matrix. The dislocation density considerably increases after the deformation.

## 4. Discussion

The microstructural observation above clearly indicates that a bcc-to-hcp phase transformation occurs in the NbTiHfZr RHEA after annealing. After cold-rolling followed by 1073 K and 1273 K-annealing, the hcp phase is precipitated. The volume fraction and size of the hcp phase increase with increasing the annealing temperature, but the lattice constant and the composition of the hcp precipitates remain unchanged. These results suggest that the hcp phase is relatively stable at the temperature range of 1073–1273 K, and the increase of the annealing temperature can precipitate more

hcp phase without grain coarsening. The hcp precipitates in the annealed RHEAs here distribute inside grains and at the grain boundaries, which is similar to the distribution of the transformed bcc precipitates at the 1473 K-annealed HfNbTiTaZr RHEA [23].

The hcp precipitates in the annealed NbTiHfZr RHEAs may be derived from the crystal structure of the constituent metals [24]. At high temperatures, the high configurational-entropy effect stables a single phase solid-solution, as evidenced by the single bcc microstructure in the NbTiHfZr RHEA after a homogenization at 1573 K [16]. With decreasing the temperature, the configurational-entropy effect is not high enough to suppress the enthalpy effect. Thus, a phase transformation occurs. Nb is a bcc structure at low and high temperatures. For Hf, Zr and Ti, the hcp structure is the stable phase at low temperature, and the bcc structure is stable at high temperature. Their binary phase-diagrams show a bcc-to-hcp transformation at high temperatures [24]. Hence, it's very possible for the NbTiHfZr RHEA to show hcp precipitation as seen in the binary phase-diagrams.

The hcp precipitates can simultaneously enhance the strength and ductility of the RHEA. Compared with that of the as-cast RHEA, the strength-ductility property of the cold-rolled RHEA is enhanced after the 1073 K annealing treatment, and further enhanced after the 1273 K treatment. It is noteworthy that, although the grain size coarsens by 10 times with increasing temperature from 1073 K to 1273 K, the yield strength of the 1273 K-annealed RHEA is almost the same to that of the 1073 K-annealed RHEA, and the uniform elongation improves by ~ 6%. Such enhancement of strength-ductility can be attributed to the increment of the volume fraction of the hcp nano-precipitates. It is clear that, in the 1273 K-annealed RHEA, the hcp precipitate is 10 times larger than that in the 1073 K-annealed RHEA. The hcp precipitates are effectively obstacles for the dislocation movement. When the dislocations move around the nano-precipitates, dislocations can bow and by-pass the precipitates to continuously move, which leads to a dislocation pile-up around the hcp precipitates, as shown in Figure S8 in the Supplementary Materials, and thus improves the strength and strain-hardening of the annealed RHEA [32]. This enhancement of strength and ductility in RHEAs via increasing hcp nano-precipitates is unique. Without the hcp precipitates, the strength-ductility of the single bcc phase RHEAs must be degraded with coarsening the grains [33].

## 5. Conclusions

Our above results highlight that hcp precipitation occurs in the annealed NbTiHfZr RHEA, which was reported to have a single bcc phase microstructure. Post-deformation annealing at 1073–1273 K facilitates the bcc phase transform to the hcp precipitates, and causes the volume fraction of the hcp precipitates to be increased with increasing annealing temperature. The hcp nano-precipitates can simultaneously enhance strength and ductility, which has never been reported in previous studies. It is clear that in RHEAs, the precipitates not only will not do harmful effect, but also can enhance mechanical performance. This can strengthen the service security when using RHEAs at high temperatures. The enhancement is tailorable as a function of the volume fraction of the hcp nano-precipitate, indicating that with the high volume fraction of the hcp nano-precipitate, the strength-ductility property can be further enhanced via adjusting post-deformation heat-treatment methods (annealing temperature, time, etc.). This may provide a new strategy by utilizing metastability at high-temperature to design high strength and ductile RHEAs.

**Supplementary Materials:** The following are available online at http://www.mdpi.com/2075-4701/9/5/485/s1, Figure S1. The final rolled-sheet (the total thickness reduction is ~ 77.8%). Figure S2. Figure S2. The binary white-and-black images from the EBSD images. (a) As-cast. (b) 1073 K. (c) 1273 K. The red points in EBSD images are transformed to the black points in the binary images. We measure the areas covered by black points in each binary image from the MATLAB based on grayscale values. The grain boundaries covered by black lines are excepted. The calculated area fractions, shown in the binary images, are corresponding to the volume fractions of hcp phases for three RHEAs. Figure S3. Energy-dispersive spectroscopy (EDS) point analysis revealing the chemical composition of the hcp precipitates. (a) Scanning TEM high-angle angular DF (STEM-HAADF) image of a scanning area with hcp precipitates in the 1073 K-annealed RHEA. At one of them with large size EDS point test is performed. (b) Point analysis result showing the hcp precipitate is rich in Hf and Zr, poor in Nb and Ti. Figure S4: Corresponding plots of strain-hardening rate versus true strain. Figure S5: Fracture morphologies of (a1-a3)

as-cast alloy; (b1-b3) cold rolled alloy and annealed at 1073 K; (c1-c3) cold rolled alloy and annealed at 1273 K with different magnifications. Figure S6. Statistical analysis of the hcp precipitates in the 1073 and 1273 K-annealed RHEAs. (a) The hcp precipitates of 1073 K-annealed RHEA; (b) the hcp precipitates of 1273 K-annealed RHEA. The sizes of the hcp precipitates are measured by line-intersection method from TEM images. Size distribution of the hcp precipitates in (c) the 1073 K-annealed RHEA, and (d) the 1273 K-annealed RHEA. The red curves were obtained by Gauss fitting. The fitted average sizes of the hcp precipitates in the 1073 and 1273 K-annealed RHEAs are 153±13 nm and 174±4 nm, respectively. Figure S7. Number densities and average sizes of the hcp precipitates in the 1073 and 1273 K-annealed RHEAs. The number densities are calculated by the precipitate number normalized by the observed area in TEM images. Figure S8. TEM observations of 1273 K-annealed HEA. (a) TEM bright-filed image of the second phase without tension deformation, the inset is the selected area diffraction of matrix and second phase; (b) The high-resolution TEM (HRTEM) image; (c) HRTEM image of the matrix and the hcp phase. Red and yellow squares represent the matrix and the hcp phases respectively; (d) Bright-filed image of the hcp particle after plastic deformation. The hcp phase is surrounded by dense dislocation plug. The inset further shows the dense dislocation plug.

**Author Contributions:** Data curation, Y.M., Y.J., P.H., Y.B., J.L. and H.W.; formal analysis, S.W., X.C. and G.W.; funding acquisition, X.C. and G.W.; investigation, G.W.; methodology, G.W.; project administration, X.C.; supervision, X.C., G.W. and Q.Z.; writing—original draft, Y.M.; writing—review and editing, Y.M.

**Funding:** This research received no external funding.

**Acknowledgments:** This project is supported by the National Key Basic Research Program from MOST (No. 2015CB856800), the National Natural Science Foundation of China (51761135125 and 51671120), the Materials Genome Engineering of Rare and Precious Metals in Yunnan (2018ZE007), and the 111 project (No. D16002).

**Conflicts of Interest:** The authors declare no conflict of interest.

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
