# Peer review of "Hexagonal Closed-Packed Precipitation Enhancement in a NbTiHfZr Refractory High-Entropy Alloy"

_metals, doi:10.3390/met9050485_

Round 1

Reviewer 1 Report

See attached file.

Author Response

Dear reviewer 1,

We would like to thank you for giving us a chance to resubmit the paper, and also thank the reviewers for giving us constructive suggestions which would help us in depth to improve the quality of the paper. Here, we present our responses to the comments of the manuscript entitled “Hexagonal closed-packed precipitation enhancement in a NbTiHfZr refractory high-entropy alloy”.

For detailed replies, please refer to the attachment.

Reviewer 2 Report

The paper presents very interesting findings, where the ductility and strength of HfNbTiZr were mutually improved by cold-rolling and subsequent heat treatments.

The manuscript needs major changes. The authors cannot claim that the HCP nano-precipitates are responsible for the increase in strength and ductility because their amount was incorrectly measured. The authors should quantify the amount of HCP nano-precipitates precisely to claim that the precipitation of HCP is responsible for such increase in properties. Any assumptions on the effects in mechanical properties based in the amount of HCP phase is inaccurate.

The authors should also make a better effort on improving the introduction as well as the English in the manuscript.

Some comments and remarks below.

Introduction

1 - line 26 – The authors should improve the sentence “a group of materials with the melting temperatures (Tm 2123 K)” – e.g. a group of materials with the melting temperatures (Tm) 2123 K.

2 - Line 27-28 – The authors should correct the sentence “the development of high-entropy alloys (HEAs) that are used at high temperatures” – HEAs are currently not used, they might have potential to be developed to be used in the future.

3 - Line 36 – It is not clear which reference belongs to each proposed ductilization method. Please, put the references after each method.

4 - Line 36 – On the quote “reducing the sample size”: one cannot improve the ductility of a material simply by reducing the size of the specimen. It might give a higher apparent ductility in micro/nanoscale (e.g. micropillars), but it is not a real improvement in the materials plasticity and ductility when considering the usual engineering structural applications. As suggested by [11], the plasticity is size-dependent and therefore can be explored for applications in micro/nanoscale. The authors can mention this finding on [11] in a different approach, not as a ductilization method.

5 - Line 37 – “were used to ductilizing HEAs [11-16]” – The authors should be more precise: were proposed.  Some of the references are interesting, but are not an attempt to ductilize RHEAs – some are just reports of the properties of RHEAs, such as in references [13] and [14].

6 – Line 44 –we apply a simple methodology to improve the ductility of a bcc HEA”: please, specify here what is the methodology.

Materials and Methods

7 – Line 58 - The authors claimed to have 77.8 % cold reduction. Please give a more comprehensive explanation of cold rolling procedure, since the alloy is BCC+HCP and expected not to be so easily rolled. (Any consequences of the rolling, presence of any cracking propagation, if there were any heating steps, etc).

Results

8 - Line 83 – Please, clarify if the grain size distribution was measured also by EBSD or by other method.

9 - Line 85 – “growing to 140 - 150 μm” What is the error of the grain size? Please, specify.

10 - Line 90-94 – What step size was used in the EBSD? If the precipitates are – as claimed by the authors – 100 nm in size, it is not possible to accurately measure the amount of HCP phase by EBSD. The measurement is incorrect and it is not possible to accurately detect 0.03%, 0.41 and 1.76% of HCP phase. Please provide a proof of these values, as EBSD does not possess sufficient resolution for nanoprecipitation indexing, especially on such low magnification. The authors show that there is some very small amount of HCP in annealed conditions from XRD and from TEM, but they cannot say the % vol.

11 - In figure 1, what are the areas with zero solutions in the EBSD images (non-indexed white areas)? Is it a third phase? Oxides? Please, clarify. The red dots showing the supposedly HCP precipitates are not a real representation of the amount of HCP phase in the material. The HCP phase can even be a result of wrong indexing. It might even not be there – as it was further mentioned on the line 178 for the as-cast state “The TEM results indicate that only the bcc single phase can be observed. The hcp phase that observed by the EBSD is not found”. Please, quantify the precipitation properly. Why the scale changes from 300 μm (figures (e) and (g)) to 30 μm (figure(f))?

12 - Line 99-101 – Please, clarify if the mentioned composition is from the precipitation region only (where it is depleted in Ti and Hf-rich), or if it is considering the whole mapping area of figure 2.a. In case the calculated average composition is from the whole area, one cannot use such high magnification in a very small area to quantify the whole bulk, also one cannot infer that the HEA is equiatomic in case of the same average composition in a bigger area - low magnification.

13 - Line 101-102 – “In the present study, Nb is the element with 102 the highest melting temperature (2750 K) followed by Hf (2506 K), Zr (2128 K) and Ti (1941 K)”. Please, elaborate more on how it is related to the phase separation.

14 - In figure 2, please specify if the Hf-rich, (Ti,Nb)-depleted areas correspond to the HCP precipitates.

15 – In Table 1, it is not clear whether the composition was measured by STEM-EDX and if the composition showed is from the bulk or if it is from the HCP precipitates. In case it is from the precipitates, how the composition of the HCP precipitates was calculated (point analysis, mapping)? How the composition of the nano-precipitates in the as-cast state was measured if it was claimed to not be present (line 178)?

16 – Lines 155 – 157 –  The authors claim that: "The increase in the strain hardening rate in region II can be attributed to the activation of martensitic transformation in the early plastic stage as analogous to the case in titanium alloys " based on strain hardening rate changes. However, there is no evidence supporting this claim shown in the microstructure- please delete it, since the microstructure after tensile deformation is not comprehensively evaluated. The HCP phase formed by transformation during tensile testing would have a distinctive shape, which is not presented anywhere. 

17 - Line 178 – “The TEM results indicate that only the bcc single phase can be observed. The hcp phase that observed by the EBSD is not found”. It was incorrectly indexed in EBSD. One cannot see 100 nm precipitates in EBSD without having a powerful SEM using extremely low step size. Clearly it was wrongly measured from the figure 1.

18 - Line 179 – “The hcp phase that observed by the EBSD”. Please correct: The hcp phase that was observed by the EBSD.

19 - Line 182 – typo: “microstructure”. Please, correct it.

20 - Figure 4.e – Please, specify which square colors correspond to which phase.

21 - Line 185 – Please, clarify how the average particle size of the HCP was determined to be an average representation of the HCP particles.

22 – Line 189 – “The increased dislocation density is responsible for the increase of the HEA plasticity” – this claim is very controversial, as the authors did not provide any calculation on dislocation density. The authors need more results to prove the statement on this case, as the increase in plasticity can be attributed to some other effect. It is usually the other way around (increase in dislocation density decreases the plasticity because it is increasing the yield strength).

Discussion

23 – Lines 204-206 – Please, prove that you really have HCP precipitates in the as-cast state before making such statement.

24 – Lines 209–210 – “the volume fraction of the hcp precipitates is almost ten or one hundred times larger than that of the as-cast HEA”. Please, clarify how the percentage of the HCP phase was accurately calculated to be able to claim such statement.

25 – Line 212-213 the increase of the annealing temperature can precipitate more hcp phase without grain coarsening.” Please provide proof of the increased amount of HCP volume fraction with increasing the annealing temperature. The authors do not provide an accurate volume fraction of HCP precipitates in any condition.

Conclusions

26 - Line 242 – “The thermal-mechanical process, i.e. cold-rolling followed by annealing at 1073 ~ 1273 K, facilitates the bcc phase transform to the hcp precipitates” – the claim that the HCP precipitation is facilitated due to the thermo-mechanical process is not proved, as it could be simply triggered due to subsequent heat treatments – not necessarily related to the cold rolling process.

27 - A CALPHAD calculation by e.g. Thermocalc software would have been proved useful for a better understanding of the phase diagram in order to check for HCP precipitation – although it is not necessarily required.

Author Response

Dear Ms. Harley Wang and reviewers,

We would like to thank you for giving us a chance to resubmit the paper, and also thank the reviewers for giving us constructive suggestions which would help us in depth to improve the quality of the paper. Here, we present our responses to the comments of the manuscript entitled “Hexagonal closed-packed precipitation enhancement in a NbTiHfZr refractory high-entropy alloy” 

For detailed replies, please refer to the attachment.

Round 2

Reviewer 1 Report

The authors have answered my questions in a conclusive fashion. The novelty content of this work remains relatively low. I anyhow recommend it for publication in the current form.

Reviewer 2 Report

The authors answered well all the reviewers remarks and and improved quality of the manuscript. The paper appears to be ready for publication.